# Liaison psychiatry—measurement and evaluation of service types, referral patterns and outcomes (LP-MAESTRO): a protocol

Chris Smith,[1] Jenny Hewison,[1] Robert M West,[1] Elspeth Guthrie ,[1] Peter Trigwell,[2] Mike J Crawford,[3,4] Carolyn J Czoski Murray,[1] Matt Fossey,[5] Claire Hulme,[6] Sandy Tubeuf,[7] Allan House [1]

For numbered affiliations see end of article.

**Correspondence to**
Dr Chris Smith;
C.J.Smith@leeds.ac.uk

## ABSTRACT

**Introduction** We describe the protocol for a project that will use linkage of routinely collected NHS data to answer a question about the nature and effectiveness of liaison psychiatry services in acute hospitals in England.

**Methods and analysis** The project will use three data sources: (1) Hospital Episode Statistics (HES), a database controlled by NHS Digital that contains patient data relating to emergency department (ED), inpatient and outpatient episodes at hospitals in England; (2) ResearchOne, a research database controlled by The Phoenix Partnership (TPP) that contains patient data relating to primary care provided by organisations using the SystmOne clinical information system and (3) clinical databases controlled by mental health trusts that contain patient data relating to care provided by liaison psychiatry services. We will link patient data from these sources to construct care pathways for patients who have been admitted to a particular hospital and determine those patients who have been seen by a liaison psychiatry service during their admission.

Patient care pathways will form the basis of a matched cohort design to test the effectiveness of liaison intervention. We will combine healthcare utilisation within care pathways using cost figures from national databases. We will compare the cost of each care pathway and the impact of a broad set of health-related outcomes to obtain preliminary estimates of cost-effectiveness for liaison psychiatry services. We will carry out an exploratory incremental cost-effectiveness analysis from a whole system perspective.

**Ethics and dissemination** Individual patient consent will not be feasible for this study. Favourable ethical opinion has been obtained from the NHS Research Ethics Committee (North of Scotland) (REF: 16/NS/0025) for Work Stream 2 (phase 1) of the Liaison psychiatry—measurement and evaluation of service types, referral patterns and outcomes study. The Confidentiality Advisory Group at the Health Research Authority determined that Section 251 approval under Regulation 5 of the Health Service (Control of Patient Information) Regulations 2002 was not required for the study 'on the basis that there is no disclosure of patient identifiable data without consent' (REF: 16/CAG/0037).

## Strengths and limitations of this study

► Study designs based on the linkage of routinely collected NHS data enable the generation of evidence regarding the cost-effectiveness and efficiency of health services, where alternative study designs such as randomised controlled trails or other individually consented study designs would be infeasible.

► While our study will evaluate the impact of acute inpatient hospital work carried out by liaison psychiatry teams and will not cover work undertaken in the emergency department, outpatient or primary care setting, the findings will be highly relevant, as previous claims for cost savings resulting from the implementation of liaison psychiatry services have been primarily based on their inpatient hospital work.

► Study designs based on the linkage of routinely collected NHS data present technical, ethical and legal challenges which must be considered from the start, including their implications for the validity and generalisability of insights and for project resources.

Results of the study will be published in academic journals in health services research and mental health. Details of the study methodology will also be published in an academic journal. Discussion papers will be authored for health service commissioners.

## INTRODUCTION

Liaison psychiatry services provide assessment and treatment for people with coexistent physical and mental health problems.[1–4] Such services are provided predominantly in the acute hospital setting in the UK, although more recently services have emerged to support the management of people with complex physical and mental health problems in primary as well as secondary care.[5] Liaison psychiatry services have the potential to improve both the quality of care and overall outcomes for people with mental and physical health problems. There is also

BMJ

a suggestion that liaison psychiatry services in the acute hospital setting will produce cost savings by reducing length of stay, even though it is estimated that services see a small proportion (1%–5%) of all patients admitted to acute beds.[6–12] For these reasons, NHS England has invested in expanding liaison psychiatry services to acute hospitals.[13 14]

The research evidence has not been strong for the cost-effectiveness of liaison psychiatry *services*[15 16] as opposed to evidence on the cost-effectiveness of some of the *individual interventions* used in those services.[17 18] For that reason, more research is needed using robust designs derived from health services research. Claims for cost-effectiveness of liaison psychiatry go to the origins of the specialty before World War II. Individual components have been tested in randomised controlled trials (RCTs), but there have been few attempts to judge the cost-effectiveness of whole services. Although it has been argued that they can pay for themselves—the cost-offset debate of the 1980 and 1990s[19]—in truth, the cost-effectiveness evidence for any liaison psychiatry service is limited. Holmes *et al*[20] identified only two RCTs; some smaller non-randomised studies include working age and older patients, and older and non-definitive work on cost-offsetting.

There are three main reasons why visiting the question now is timely.

First, cost pressures in the NHS and the emergence of commissioning led by Clinical Commissioning Groups will continue to lead to re-profiling of services and especially attempts to reduce unnecessary hospital costs. Work on developing and implementing risk stratification models is an illustration and it is interesting that many of these models identify comorbid physical and mental health problems as a risk, both for increased healthcare costs and for the main driver of such costs which is unscheduled hospital admission. A study from Birmingham describing the evaluation of a rapid assessment, intervention and discharge service (previously widely known by the acronym RAID) was widely promoted and is leading to commissioning of similar services that will be hoped to reduce costs.[21 22] There were however substantial problems with that evaluation: it reported only on the first 9 months of delivery of a new service; it used a simple before and after design; it compared outcomes between referred and a matched group of non-referred patients in only 79 cases with minimal matching that cannot have dealt with indication bias and much of the benefit was attributed to the so-called indirect liaison cases who were in fact not seen by the service but assumed to benefit by its existence in the hospital. Other 'RAID'-like services have also reported large savings in cost or reductions in hospital use following implementation.[23 24] So a key answer to the question of why this research is needed now is the pressing need to confirm or refute the very striking claims made for similar services, but using larger numbers and more robust research methods.

Second, there is a danger of losing sight of the other main functions of liaison psychiatry services, which do not exist only to reduce costs in the general hospital but which are aimed at improving the well-being of patients, some of whom are being treated entirely or appropriately in the general hospital and happen to need mental health service input because of the complexity of their problems.

Third, and related to the second point above, service commissioning and provision will benefit from a more standardised approach to service descriptors. Without more detailed knowledge about how to define the service being commissioned and how to evaluate whether it is working to remit and to improve Health-Related Quality of Life for patients, there is a risk of enthusiastic commissioning of services that look superficially similar to each other and to a model reported as cost-effective, when in fact either at the time or over a period after commissioning there are differences in staffing or referral patterns that invalidate the original commissioning assumptions. An important function is served by studies that describe what patient groups actually receive—what sort of service and in what numbers.

This research is thus timely in exploring methods to evaluate the function and performance of liaison psychiatry services.

There are, however, challenges in conducting such research.

First, defining exposure to liaison psychiatry is difficult because there is substantial heterogeneity in the composition, purpose and size of liaison psychiatry services. For example, a recent survey conducted in England[25] found that just over half of the services provided a 24-hour 7-day service and only one-third ran specialist outpatient clinics. Most of the services provided cover of acute hospital wards and emergency department (ED) and nearly all services were multidisciplinary, but staff mix varied such that about one-third employed less than one full-time equivalent of a consultant in psychiatry. Only one-third of services had separate teams for older adults and adults of working age. These differences were not fully explained by variation in acute hospital characteristics. Also, the mechanism by which liaison psychiatry services might produce improved patient and organisational outcomes is unclear—for example, with some suggesting that the main benefit is by securing rapid discharge to community-based services, while others emphasise the specialist nature of shared inpatient management or of outpatient clinics provided by the service.

Second, defining service outcomes is difficult because there is also substantial heterogeneity in the patient populations seen in liaison psychiatry services. For example, service outcomes (and performance indicators) will not be the same for somebody seen in the ED after an act of self-harm, an older person with postoperative agitated behaviour seen on a surgical ward or somebody with chronic unexplained pain referred from a pain clinic.

To evaluate the impact of liaison psychiatry services on the outcomes of patients in acute hospital settings, we therefore need to be able to do three things. *Patients* attending selected hospitals need to be characterised with respect to their physical and mental health. The prognosis for any given mental health problem is strongly influenced by the prior history of mental disorder, so the nature of the psychiatric problem needs to be described not just at the time of admission but in the preceding months. This longitudinal picture can only be determined reliably and for all patients in any sample by the use of routine data from primary care. Patient *healthcare contact* in both primary and secondary care services needs to be recorded. And *outcomes* need to be identified beyond the immediate spell in hospital. The heterogeneity of both patient population and service exposures requires a large study in terms of number of hospitals and patients.

In addition to these measurement problems, there is a challenge in choosing a robust research design. There have been several RCTs showing the effectiveness of individual components of liaison psychiatry services, but an individually randomised RCT of a whole service configuration would be impractical. The *heterogeneity of case mix* even in simpler services will require numbers beyond what could be reasonably recruited, and there are major challenges in obtaining a large representative sample when individual consent is required. Because many patients seen by liaison psychiatry services lack mental capacity at the time of service contact, an individually consented study is not feasible and an individually RCT study would not in any case answer the service-level question.

We also considered a cluster RCT of different liaison psychiatry services. *Heterogeneity of service provision*, as identified in our national survey of services in England, would make such a study prohibitively large.

For these reasons, our view is that a large-scale observational study based on analysis of *routinely collected NHS data* and which is not predicated on individual patient consent is the best option.

## METHODS AND ANALYSIS
### Study design
Retrospective cohort design, comparing outcomes for patients admitted to hospital and seen by a liaison psychiatry service with two control groups—the first is a patient group who were admitted to the same hospital in the same study period and matched on hospital inpatient and primary care data, but were not seen by the liaison psychiatry service. Because such a design could not entirely exclude confounding by indication, we will use a second matched patient group who had been admitted to a different hospital without a liaison psychiatry service in the same study period. This second group will also be matched using data from hospital and primary care records; however, they will not have been selected on the basis of whether the responsible (acute hospital) consultant had made a decision about liaison psychiatry referral.

### Aims and objectives
This study arises from a commissioning call by the UK's National Institute for Health Research and represents one component (Work Stream 2) of the wider project, LP-MAESTRO,[26] which is designed to evaluate the cost-effectiveness and efficiency of liaison psychiatry service provision in the UK. The aim of the study described in this paper is to examine care pathways for the main target populations of liaison psychiatry services and estimate the outcomes and costs associated with care. Specifically, we will:

1. Use routinely collected NHS data to identify patients referred to specific liaison psychiatry services and matched comparison patients who were not referred, with the aim of comparing within and between hospitals the effect of referral or non-referral of patients with similar characteristics.
2. Estimate the cost of the care pathways of patients referred to liaison psychiatry services, and the matched comparison patient groups, and the main determinants of those costs over 12 months after an index hospital episode.

We will characterise patients according to their contact with liaison psychiatry service, for example reason for referral, scheduled or urgent referral and mental health diagnosis. We will determine those patients who were referred to liaison psychiatry services from acute (general hospital) sources, and a matched sample of cases from the same sources who were not referred. We will compare outcomes for certain marker conditions (such as mental–physical comorbidity, acute behavioural disturbance, cognitive impairment/dementia) in different liaison psychiatry service configurations.

### Data sources
Patient data that are relevant to the study are routinely collected by clinicians and healthcare professionals to inform patient care. Such data are collected independently by the organisations that provide different services to patients, and only those variables that are required to fulfil the purpose(s) of these services are included. A limited number of standardised data sets are collected from organisations that provide care and are aggregated at national-level by organisations, such as NHS Digital. Some of these data sets (or derivatives) are made available for research purposes. However, no single organisation can currently provide the data that are required for the study, so linkage is essential.

Hospital Episode Statistics (HES)[27] is a database controlled by NHS Digital that contains patient data relating to ED, inpatient and outpatient episodes for NHS hospitals in England. Episodes represent discrete periods of care under a particular consultant. Episodes can be combined into spells to represent the period of care from admission to discharge. HES is derived from the Commissioning Data Set,[28] which is supplied to NHS Digital by organisations that provide NHS services to facilitate monitoring and payment. Patients can opt-out

of the inclusion of their *confidential patient information* in data sets which are made available by NHS Digital for purposes beyond care, such as HES, through the national opt-out programme.[29] HES is an important source of data relating to health service interaction in secondary care. There are three significant limitations that are relevant to this study.

First, referral to liaison psychiatry services cannot be reliably determined from HES. A new episode is not generated by such a referral—the patient remains within the care of the acute hospital consultant—and contact with a different consultant-led team in liaison psychiatry is not represented within an episode.

Second, it is suggested that mental health diagnoses recorded in routinely collected NHS data, such as HES, exhibit variable accuracy with respect to the true diagnosis.[30]

Third, patient interactions with primary care services are not included in HES. Such data are required to match patients by defined characteristics and to determine the care delivered in primary care following a general hospital admission that leads to a liaison psychiatry referral.

To address the first limitation, clinical databases controlled by the mental health trusts that provide liaison psychiatry services will be used. Such databases contain patient data relating to care provided by liaison psychiatry services and can be used in conjunction with HES data to determine whether a patient was referred to a liaison psychiatry service during a hospital admission. The main challenges with the use of such databases are data quality and the heterogeneous processes by which data access is negotiated and administered within different organisations.

To address the second and third limitation, data relating to primary care is required for each patient. Although the Clinical Practice Research Datalink database[31] is widely used for primary care research in the UK, it has a major drawback for this study—limited numbers and geographical coverage of participating primary care organisations at the time of study design. Instead ResearchOne,[32] a research database controlled by The Phoenix Partnership (TPP), will be used. ResearchOne contains patient data relating to primary care provided by organisations using the SystmOne clinical information system.[33] SystmOne (34%) and EMIS[34] (56%) are the most prevalent clinical information systems used by organisations in general practice.[35] Therefore, many of the patients with episodes in the HES data can be expected to have data relating to their primary care collected by organisations that use SystmOne. ResearchOne contains data for a subset of these patients—patients who: (1) are registered to organisations that use SystmOne and have opted-in to participation in ResearchOne and (2) have not individually opted-out of participation in ResearchOne. The main challenge with the use of ResearchOne is the inability to determine a priori the numbers and geographical coverage of organisations that provide primary care to patients attending the hospitals chosen for the study, and

the resultant ability to match HES data to corresponding ResearchOne data for each patient.

Patient data from these three sources will be linked to construct patient care pathways that span primary and secondary care settings. Linkage will be undertaken by NHS Digital. Each data source will generate two unique references for each patient: (1) a pseudonym—generated by applying a one-way cryptographic hash function (SHA-512) to an input that comprises a cryptographic salt and their NHS number and (2) a source-specific identifier. For a patient with a given NHS number, each data source will generate the same pseudonym but a different source-specific identifier. Both the pseudonym and source-specific identifier generated for each patient will be specific to the study. Pseudonyms will be used by NHS Digital to: (1) communicate to data sources of those patients for whom data are required and (2) generate mappings between different source-specific identifiers for each patient. Data sources will provide the required patient data to the research team, including only the source-specific identifier as the unique reference for each patient. The mappings generated by NHS Digital will be provided to the research team and used to determine the data that relate to each patient across the data received from different sources.

### Data extraction

Based on the results from earlier stages of the LP MAESTRO project, we will identify at least two and up to six configurations that best represent patterns of liaison psychiatry service across England. Defining features of such configurations will include, for example, staff mix, availability of specialist teams (eg, age-related, self-harm), hours of service provided by the specialist team and source of referrals (predominantly ED, predominantly ward, specialist services and so on). We will sample purposively to obtain 2–4 services of each type (depending on availability). Data will be extracted for patients attending the hospitals with these service elements in a 1-year index period and also for patients attending hospitals identified as not having a liaison psychiatry service in the same period. Financial year 2013–2014 was selected as the index period as it represented the latest complete year for which data was available from NHS Digital at the time of study design. Numbers of accident and emergency (A&E) attendances and inpatient admissions in the index period to estimate sample size are not publicly available at hospital level. However, numbers are published by NHS Digital at trust level,[36 37] where one trust operates one or more hospitals. These trust-level figures provide an indicative upper bound on the A&E attendances and inpatient admissions that can be expected at any hospital operated by that trust in a given year.

Relevant variables extracted for each patient from the sources will include demographic variables (eg, age, carer support, index of multiple deprivation—a measure of locality deprivation), clinical variables (eg, diagnosis, medications) and health service utilisation variables

(eg, inpatient days, general practitioner (GP) appointments, major procedures). One of the novelties of our approach is the use of variables obtained from primary and secondary care settings to tackle the substantial challenge that comes from indication bias; for example, we will use variables obtained from ResearchOne to define healthcare usage in primary care for the year before referral (2013), as a way of ensuring that outcomes in the year after referral (2015) are not attributed to easily identifiable pre-existing characteristics (case complexity) that are confounded with likelihood of referral.

Patient care pathways for patients attending hospitals using liaison psychiatry services in each configuration will be constructed to provide a view of health and healthcare across both primary and secondary care. Pathways will be constructed for patients for a period of 12 months following their index (first) hospital admission in the index period. The cost of each pathway to the healthcare sector will then be calculated using national data sources (see below). We will adopt a whole system perspective in order to determine if there is an association between the configuration of liaison psychiatry services and healthcare utilisation by patients. Metrics including emergency admissions, occupied bed days and length of stay will be analysed by age band.

## Data analysis

We will build a standard regression model to estimate the relation between healthcare utilisation and key variables capturing the configuration of liaison psychiatry services. The dependent variable in this model will therefore be the total costs of any identified healthcare utilisation derived from factors such as inpatient days, readmission rates, ED attendances and GP visits combined with reference costs. These reference costs will be valued using national sources, including the Department of Health Reference Costs[38] and Personal Social Services Research Institute Costs for Health and Social Care.[39] Where these are not available, local costs will be assigned. We will choose the most appropriate base year for the analysis and adjust appropriately for the effects of inflation across years. The quantum of the liaison psychiatry service provision will be captured by already collected data related to structure and process, for example, staffing levels and contact time after referral. We will adjust for referral indication bias, either by matching for covariates or by propensity scoring.

Sample size is difficult to estimate because we have so little available data on outcomes for different service types and different patient groups. Suppose we identify six main service configurations and recruit two liaison psychiatry services for each (total n=12). For less common conditions we might expect to see one referral per week per service=100 in total in the year. For more common conditions we might perhaps see one referral per day or 600–800 in the year. These numbers will allow us to estimate the costs and cost-effectiveness of liaison psychiatry services with substantially greater precision than has been achieved to date, for example, by the RAID evaluation.

The way in which components of general hospital, general mental health and liaison psychiatry services interact with each other is complex and a key part of the project will be to determine how to capture this complexity into a set of measures for inclusion in the model.

We (CH, ST) will carry out exploratory incremental cost-effectiveness analyses using decision analysis modelling. The model will rely on the retrospectively estimated healthcare costs of alternative care pathways and the characteristics at hospital, service and patient level. Given the nature of the data available, the absence of measures such as Quality of Life measures and the heterogeneous nature of the population, we will explore the use of a range of variables to assess effectiveness and evaluate the costs per length of stay, per re-admission and per life years lost. The health economics analyses will be informed by earlier work packages in the Liaison psychiatry—measurement and evaluation of service types, referral patterns and outcomes (LP-MAESTRO) project. However, we will also follow the guidance from the National Institute for Clinical Excellence (NICE) for methods for technology appraisal.[40] While it is clear that some aspects of an exploratory model (or indeed models) may be specified in advance, for example, the perspective of the economic evaluation which will be the health service provider and the comparator which will be usual care, other aspects will be dependent on the shape of the services and the populations they engage with.

At this point we are unable to specify the time horizon of the decision analysis model evaluating the long-term cost-effectiveness of liaison psychiatry services. We will look to a long-term model and use NICE recommended discount rates for costs. The model itself is likely to be Markov or semi-Markov. Sensitivity analyses will be undertaken in line with those recommended for this type of modelling.[41] Presentation of the analysis or analyses will include incremental cost-effectiveness ratios, cost-effectiveness acceptability curves and net monetary benefit estimates. In addition, we will undertake a value of information analysis.[42]

## Patient and public involvement

The study has a patient and public involvement representative on the Study Steering Committee that oversees the management of the research.

## ETHICS AND DISSEMINATION

The study is funded by the National Institute of Health Research under the Health Services and Delivery Research programme (REF: 13/58/08) and is sponsored by the University of Leeds. The study is based at the Leeds Institute of Health Sciences within the University of Leeds and will use the Information Governance Toolkit (IGT) compliant infrastructure at the Leeds Institute of Clinical Trials Research (REF: ECC0010).

To simplify the documentation provided to underpin ethics and governance processes, the study has been

partitioned into distinct phases. Each phase is characterised by the use of a specific combination of data sources to construct care pathways for patients attending a particular hospital or set of hospitals. Phase 1 is characterised by the use of data from HES and ResearchOne *only* to construct care pathways for patients attending hospitals *without* a liaison psychiatry service. A summary of the ethics and governance processes undertaken for phase 1 is provided below.

As described in the Introduction section, individual patient consent will not be feasible for this study.

Phase 1 was submitted to the NHS Research Ethics Committee (North of Scotland) on 23 February 2016 (REF: 16/NS/0025). The application was reviewed in a meeting held on 10 March 2016 and received a favourable ethical opinion on 15 March 2016. Favourable ethical opinion was contingent in obtaining management permission from the hospitals whose patients were to be included in the HES data. Management permission was received for 8 of the 11 hospitals by a stated deadline (15 December 2016) and phase 1 proceeded based on the use of HES data from these eight hospitals only.

Phase 1 was also submitted to the Confidentiality Advisory Group (CAG)[43] at the Health Research Authority on 23 February 2016 (REF: 16/CAG/0037) to determine whether the study required Section 251 support under Regulation 5 of the Health Service (Control of Patient Information) Regulations 2002.[44] The application was reviewed in a meeting on 21 April 2016 and was deferred pending the receipt of further information in relation to the linkage process from the research team. Further information was supplied and CAG provided a decision on 19 July 2016 that Section 251 support was not required 'on the basis that there is no disclosure of patient identifiable data without consent'.

Based on favourable ethical opinion and a decision from CAG that Section 251 support was not required, approval in principle for phase 1 was provided by TPP on 11 October 2016. Data requests were submitted to NHS Digital on 16 December 2016 (REF: NIC-77953) and TPP on 22 February 2017. Supporting evidence documents were provided with these requests, which included confirmation of favourable ethical opinion, confirmation of the CAG decision on Section 251 support and details of the technical and organisational safeguards in place at the data controller and processors. Organisational safeguards included a Data Processing Agreement established between University of Leeds and TPP to cover the data processing activities within phase 1. The application was reviewed by the Independent Group Advising on the Release of Data (IGARD)[45] at NHS Digital on 20 March 2018. Further information was requested from the project team by IGARD, which was subsequently supplied, and a recommendation to approve the application was provided in a meeting on 26 April 2018. A Data Sharing Agreement was established between the University of Leeds and NHS Digital for phase 1 on 26 April 2018, which was underpinned by the pre-existing Data Sharing Framework Contact between the University of Leeds and NHS Digital (REF: CON-315426-K3W7R).

Data were supplied by NHS Digital on 16 November 2018 and the remaining data from NHS Digital and TPP are currently awaited.

Results of the study will be published in academic journals in health services research and mental health. Details of the study methodology will also be published in an academic journal. Discussion papers will be authored for health service commissioners.

## DISCUSSION

Studies based on the linkage of routinely collected NHS data enable the generation of evidence regarding the cost-effectiveness and efficiency of health services, where alternative study designs such as RCTs or other individually consented study designs would be infeasible. Health service commissioners can be provided with robust evidence to underpin decisions regarding health services and interventions—in this case relating to the cost-effectiveness and efficiency of different liaison psychiatry configurations for specified target populations—where previously there may have been limited or no evidence. The findings of this study will evaluate the impact of acute inpatient hospital work carried out by liaison psychiatry teams, and does not cover work undertaken in the ED, outpatient or primary care setting. The findings, however, will be highly relevant, as previous claims for cost savings resulting from the implementation of liaison psychiatry services have been primarily based on their inpatient hospital work.

Studies such as this present technical, ethical and legal challenges. Study designs should consider such challenges from the start, including their implications for the validity and generalisability of insights and for project resources. Experience from LP-MAESTRO demonstrates that significant resources are required to design and communicate a research protocol in the manner that satisfies the different project stakeholders, including research ethics committees, data controllers, regulatory bodies and data access committees. Different stakeholders are focused on their specific remit and require communication of the research protocol in accordance with that remit. Moreover, different stakeholders may comprise of decision makers from different disciplines and require the research protocol to be communicated at differing levels of abstraction to ensure adequate comprehension. This issue of communication between disciplines and the potential for misinterpretation is highlighted in a recent Nuffield Foundation report.[46]

The project described here is both technically feasible and consistent with current legislative and ethical frameworks applicable to the use of health data for research purposes. The main practical challenges reside in the communication with, negotiation between and coordination of different stakeholders as outlined earlier.

There are potential limitations to the findings: it may not be easy to derive clearly discrete configurations of service from routinely available data; although matching via primary care records will allow more precision than can be managed from routine hospital data, the data available for matching will still be limited and not standardised across records; although a multi-site study will generate substantial numbers (a typical liaison psychiatry service will make 2000+ new contacts a year), sample sizes for particular subgroups of patients may be too small for meaningful analysis.

## Author affiliations
[1]Leeds Institute of Health Sciences, School of Medicine, University of Leeds, Leeds, UK
[2]National Inpatient Centre for Psychological Medicine, Leeds and York Partnership NHS Foundation Trust, Leeds, UK
[3]Centre for Psychiatry, Department of Medicine, Imperial College London, London, UK
[4]College Centre for Quality Improvement, Royal College of Psychiatrists, London, UK
[5]Veterans and Families Institute for Military Research, Faculty of Health, Social Care and Education, Anglia Ruskin University, Chelmsford, UK
[6]University of Exeter Medical School, College of Medicine and Health, University of Exeter, Exeter, UK
[7]Institute of Health and Society, Université catholique de Louvain, Louvain-la-Neuve, Belgium

**Acknowledgements** The views expressed are those of the author(s) and not necessarily those of the National Institute for Health Research (NIHR) or the Department of Health and Social Care.

**Contributors** CS, JH, RMW, EG, PT, MJC, CJCM, MF, CH, ST and AH contributed to the design of the study protocol. AH, JH and CS authored the manuscript. CS, JH, RMW, EG, PT, MJC, CJCM, MF, CH, ST and AH contributed to the revision of the manuscript and approved the final version.

**Funding** This project is funded by the National Institute for Health Research (NIHR) Health Services and Delivery Research (HS&DR) programme (project reference 13/58/08).

**Competing interests** CS is Director of PrivacyForge Limited. All other authors declare that they have no competing interests.

**Patient consent for publication** Not required.

**Ethics approval** Individual patient consent will not be feasible for this study. Favourable ethical opinion has been obtained from the NHS Research Ethics Committee (North of Scotland) (REF: 16/NS/0025) for Work Stream 2 (phase 1) of the Liaison psychiatry—measurement and evaluation of service types, referral patterns and outcomes study. The Confidentiality Advisory Group at the Health Research Authority determined that Section 251 approval under Regulation 5 of the Health Service (Control of Patient Information) Regulations 2002 was not required for the study on the basis that there is no disclosure of patient identifiable data without consent' (REF: 16/CAG/0037).

**Provenance and peer review** Not commissioned; externally peer reviewed.

**ORCID iDs**
Elspeth Guthrie http://orcid.org/0000-0002-5834-6616
Allan House http://orcid.org/0000-0001-8721-8026

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
