## [Reviewer comments · BMJ Open]

ARTICLE DETAILS

TITLE (PROVISIONAL)	Liaison Psychiatry - Measurement and Evaluation of Service Types, Referral patterns and Outcomes - LP- MAESTRO: a protocol
AUTHORS	Smith, Chris; Hewison, Jenny; West, Robert; Guthrie, Elspeth; Trigwell, Peter; Crawford, Mike; Czoski Murray, Carolyn; Fossey, Matt; Hulme, Claire; Tubeuf, Sandy; House, Allan

VERSION 1 – REVIEW

REVIEWER	Matthew Macfarlane NSW Health Australia
REVIEW RETURNED	01-Jul-2019

GENERAL COMMENTS	Thank you for the opportunity to review this study protocol regarding the evaluation of consult-liaison psychiatry (CLP) services in the NHS. This is an important topic, and the introduction gives a good overview of the current literature (mainly with the frustrations and uncertainties about what constitutes a Key Performance Indicator for CLP, and how best to measure them), as well as the limitations of the economic benefit research done into RAID. Not being from the UK, I can't speak to the suitability of the data sources that have been picked, but the rationale given in the protocol is easy to follow and make sense for the outcome measures that are envisaged. The design of the groups - medical encounter/hospital admission in which CLP had input; similar patients from the same period without CLP input; and control for indication bias via a third group from hospitals where CLP was not available - is well-explained and is probably the best method to estimate effect of CLP input short of an RCT. The basic outcome measures are concrete (length of stay, mortality, readmission etc), are often felt to be some of the more important KPIs for CLP services, and seem appropriate. Future prospective trials would do well to include QoL measures, but that is unable to be performed here with retrospective data. The calculation of health costs is not totally clear from the description in the paper, but this seems difficult to summarise without knowing which services will be in the study. As long as this is documented well in the final paper, I don't have any concerns. The ethical approval is well-documented.
--

	Overall I agree with the conclusions that this seems a valuable area of study, is designed in a way that the research questions can be answered, is ethically clear in its approach to data (no small feat considering the number of organisations and databases involved) and able to be executed successfully. I do not have any suggestions for revision or explanation, with the caveat that my knowledge of the UK-based databases and information is poor.
--	--

REVIEWER	Paul Desan MD PhD Department of Psychiatry Yale School of Medicine New Haven, CT USA
REVIEW RETURNED	30-Jul-2019

GENERAL COMMENTS	Smith et al propose to create a dataset regarding patients admitted to NHS medical hospitals linking 3 databases related to hospital admission, primary care, and mental health care. They propose to compare patients who are seen by liaison psychiatry services with "matched" patients who were not seen, as well as with patients from other hospitals which did not have liaison psychiatry services. The outcome will be health expenditures but they also will compare hospital and ED admissions, death, and other marker outcomes. The Discussion needs to acknowledge the limitations of such a design. One limitation is identifying different interventions to compare. The crux is their hope that there are "2 to 6" different types of liaison psychiatry services for them to compare. In the USA, this study design would not generate sensible comparisons: for example, day-night services versus day only services would compare systems that still see all patients referred, with slightly less delay only for night emergencies. Hospitals in the USA "lacking" liaison psychiatry make up the lacking services out of emergency staff or community providers, and tend to be in very different treatment systems where costs are not comparable. Another limitation is that "matching" may not be feasible. Observations on patients that would assess propensity for consultation request simply are not available in the US medical system, and I doubt realistic matching is possible in the NHS system either. On the outcome side, a major limitation is the absence of a clear a priori hypothesis, and the reader is entitled to anxiety about multiple retrospectively constructed hypotheses. A statistical limitation is the effect sizes on a population level will be small: subtleties in the construction of comparison groups may generate errors as great. A final limitation is the vision of the hypothesis under test. All hospitals in the USA have access to some form of "liaison psychiatry", just as they all have access to cardiologists. The literature summary provided by the authors is outdated: the concern in the USA now is that selecting which patients will benefit from consultation is critical, and the real cost savings will result from systematic screening. This is a study that compares old fashioned models and may miss the real excitement in the field. This study is an exploratory study which will generate a vast amount of detailed data, which may inform decision making and planning, whether or not the primary hypothesis of cost offset is supported. I would support publication if the limitations of the design were more explicitly expressed in a revised draft.
---

REVIEWER	Tayyeb Tahir University Hospital of Wales Cardiff, CF14 4XW
REVIEW RETURNED	08-Sep-2019

GENERAL COMMENTS	It's a welcome publication and development for those interested in Liaison Psychiatry (LP). However, it needs some clarification which perhaps are minor changes to be considered.  1. Emergency Departments (ED) and outpatient work is key to liaison psychiatry. In particular development of LP services in ED have been most influential in overall development of services across the UK. Can this limitation be justified? 2. If the focus is on inpatients work for LP then clarity is needed on type of services that are focus. What is meant by 'study services'? 3. It would be useful to show the plan in a flow diagram to show patient and matched comparison group for estimating costs for two aims of the study. 4. What do authors mean 'We will characterise patients according to their contact...'?? 5. Authors need to clarify on the comparison of outcomes for certain marker conditions in different service configurations. 6. Information from LP MAESTRO is missing. If there is existing information from earlier workstream it needs to be outlined in the background / introduction section. This is important as it is mentioned later in data extraction. 7. Data sources seem to have been well thought with linkage by NHS Digital. Data Extraction  8. This section needs clarity 9. Results from LP MAESTRO need to be mentioned. 10. Clarity is needed on 'we will sample purposively to obtain 2-4 services of each type (depending upon availability). Does it mean that the research team does not have designated services to extract data from? 11. From previous data what 2-6 configurations are likely to identified? 12. How have the authors decided 2013-14 as the year to extract data for? Why not 2018-19? The 2013 costs will be different. How will they interpret difference to make it relevant in present time? 13. For readers ease, differentiate between variables and configurations. 14. Data from year before and year after is strength to deal with an obvious bias. 15. Sample size: There should be some idea from LP-MAESTRO for the paragraph on sample size. How many patient records will be seen? Is there a power calculation or authors are going to work on an assumption? Or as many as possible? Important aspect of study. For example self harm, delirium, depression are well recognised in general hospital and have been written about. 16. Have they calculated costs for each element? 17. On the basis of the period considered the costs will vary for 2013 from 2019-20. How will they account for that difference? 18. The paragraph on analyses of costs is confusing. It will be useful to outline what economic evaluation techniques will be used. How will QoL measures explored retrospectively.
---

	Information from LP-MAESTRO is mentioned again without giving any details. 19. This is specific health economic research. What expertise is there in the team? Or the team has sought help from statisticians with this expertise or from an organisation.
--	--

VERSION 1 – AUTHOR RESPONSE

Reviewer: 1

We thank the reviewer for the supportive comments.

In response to specific points raised:

1. The calculation of health costs is not totally clear from the description in the paper, but this seems difficult to summarise without knowing which services will be in the study.

Following questions from reviewer 3 we have now made some changes to this section, which hopefully improved the clarity.

Reviewer: 2

Please leave your comments for the authors below Smith et al propose to create a dataset regarding patients admitted to NHS medical hospitals linking 3 databases related to hospital admission, primary care, and mental health care. They propose to compare patients who are seen by liaison psychiatry services with "matched" patients who were not seen, as well as with patients from other hospitals which did not have liaison psychiatry services. The outcome will be health expenditures but they also will compare hospital and ED admissions, death, and other marker outcomes. The Discussion needs to acknowledge the limitations of such a design. One limitation is identifying different interventions to compare. The crux is their hope that there are "2 to 6" different types of liaison psychiatry services for them to compare. In the USA, this study design would not generate sensible comparisons: for example, day-night services versus day only services would compare systems that still see all patients referred, with slightly less delay only for night emergencies.

Hospitals in the USA "lacking" liaison psychiatry make up the lacking services out of emergency staff or community providers, and tend to be in very different treatment systems where costs are not comparable. Another limitation is that "matching" may not be feasible. Observations on patients that would assess propensity for consultation request simply are not available in the US medical system, and I doubt realistic matching is possible in the NHS system either. On the outcome side, a major limitation is the absence of a clear a priori hypothesis, and the reader is entitled to anxiety about multiple retrospectively constructed hypotheses. A statistical limitation is the effect sizes on a population level will be small: subtleties in the construction of comparison groups may generate errors as great.

We have added a paragraph to the discussion acknowledging these limitation, apart from the point about a priori hypothesis – this is an observational/descriptive study that is not hypothesis-testing.

A final limitation is the vision of the hypothesis under test. All hospitals in the USA have access to some form of "liaison psychiatry", just as they all have access to cardiologists. The literature summary provided by the authors is outdated: the concern in the USA now is that selecting which patients will benefit from consultation is critical, and the real cost savings will result from systematic screening. This is a study that compares old fashioned models and may miss the real excitement in the field. The protocol describes a study commissioned by the UK's National Institute for Health Research, which is designed to answer a question about services as currently configured. We have changed the text in Aims and Objectives to make this point clearer.

Reviewer: 3

We thank the reviewer for the supportive comments.

In response to specific points raised:

1. Emergency Departments (ED) and outpatient work is key to liaison psychiatry. In particular development of LP services in ED have been most influential in overall development of services across the UK. Can this limitation be justified?

We have added text to data extraction to explain that this information was not included in the relevant HES data at the time of our study design.

2. If the focus is on inpatients work for LP then clarity is needed on type of services that are focus. What is meant by 'study services'?

We have added clarifying text under data extraction.

3. It would be useful to show the plan in a flow diagram to show patient and matched comparison group for estimating costs for two aims of the study.

We decided the explanatory text was sufficiently clear.

4. What do authors mean 'We will characterise patients according to their contact...'?

We have added text to clarify.

5. Authors need to clarify on the comparison of outcomes for certain marker conditions in different service configurations.

We have added text to clarify.

6. Information from LP MAESTRO is missing. If there is existing information from earlier workstream it needs to be outlined in the background / introduction section. This is important as it is mentioned later in data extraction.

We have added text to clarify.

7. Data sources seem to have been well thought with linkage by NHS Digital.

No changes made in response to this comment.

8. This section needs clarity (Data Extraction)

We have added text to provide clarity on specific points raised for the Data Extraction section.

9. Results from LP MAESTRO need to be mentioned (Data Extraction)

As this is a protocol we do not think it appropriate to report other results.

10. Clarity is needed on 'we will sample purposively to obtain 2-4 services of each type (depending upon availability). Does it mean that the research team does not have designated services to extract data from? (Data Extraction)

We do have candidate services for data extraction but for the purposes of presenting the protocol it is not desirable to name them.

11. From previous data what 2-6 configurations are likely to be identified? (Data Extraction)

We have added text to clarify.

12. How have the authors decided 2013-14 as the year to extract data for? Why not 2018-19? The 2013 costs will be different. How will they interpret difference to make it relevant in present time? (Data Extraction)

We have explained reasons for choice of year in the text, and extended the discussion of the health economics component.

13. For readers ease, differentiate between variables and configurations. (Data Extraction)
Variables and configurations represent two separate concepts. Variables refer to data items. Configurations refer to service configurations. We consider the text to be sufficiently clear in this regard and have not made any specific changes in response to this comment.

14. Data from year before and year after is strength to deal with an obvious bias. (Data Extraction)

No changes made in response to this comment.

15. Sample size: There should be some idea from LP-MAESTRO for the paragraph on sample size. How many patient records will be seen? Is there a power calculation or authors are going to work on an assumption? Or as many as possible? Important aspect of study. For example self harm, delirium, depression are well recognised in general hospital and have been written about. (Data Extraction)

We have added a note in the discussion about limitations of sample size.

16. Have they calculated costs for each element? (Data Extraction)

After each element has been identified, we will search for a reference cost if available in the Department of Health Reference Costs (reference 38) and Personal Social Services Research Institute (PSSRU) Costs for Health and Social Care (reference 39) as we usually do for health economics studies. If a cost is not available then we will seek local experts for local cost.

17. On the basis of the period considered the costs will vary for 2013 from 2019-20. How will they account for that difference? (Data Extraction)

For the cost-effectiveness analysis, we will choose the most appropriate base year for the analysis and this could be 2018-19, in that case, we would simply adjust appropriately for the effects of inflation across years (see: Donaldson C, Shackley P. Economic studies. In: Oxford Textbook of Public Health. 3rd ed. Detels R et al. (eds.). Oxford. Oxford University Press. 1997).

18. The paragraph on analyses of costs is confusing. It will be useful to outline what economic evaluation techniques will be used. How will QoL measures be explored retrospectively. Information from LP-MAESTRO is mentioned again without giving any details. (Data Extraction)

In the absence of QoL measures, we will use other outcomes as effectiveness measures, it includes length of stay, readmission and mortality. Therefore the economic evaluation techniques will be a comparative cost-effectiveness using length of stay, readmission and life years lost as the outcomes. We have made some changes to the text to reflect this.

19. This is specific health economic research. What expertise is there in the team? Or the team has sought help from statisticians with this expertise or from an organisation. (Data Extraction)

At the time of the design of the protocol, there were 2 senior health economists involved (Prof. Claire Hulme and A/P Sandy Tubeuf), both have now left the University of Leeds but have been replaced by Prof. Chris Bojke, who has expertise in HES data and modelling of HTA and Senior Research Fellow Dan Howdon, who has expertise in large data analysis and applied econometrics methods. The health economists will work closely with the statisticians for the delivery of the research.

VERSION 2 – REVIEW

REVIEWER	Paul Desan MD PhD Dept of Psychiatry Yale School of Medicine New Haven, CT USA
REVIEW RETURNED	12-Oct-2019

GENERAL COMMENTS	The concerns of the reviewers have been addressed as far as feasible, and I support publication of this protocol.
---